# Cytosolic Isocitrate Dehydrogenase from *Arabidopsis thaliana* Is Regulated by Glutathionylation

**DOI:** 10.3390/antiox8010016

**Published:** 2019-01-08

**Authors:** Adnan Khan Niazi, Laetitia Bariat, Christophe Riondet, Christine Carapito, Amna Mhamdi, Graham Noctor, Jean-Philippe Reichheld

**Affiliations:** 1Laboratoire Génome et Développement des Plantes, Université Perpignan Via Domitia, F-66860 Perpignan, France; niazi@uaf.edu.pk (A.K.N.); laetitia.bariat@univ-perp.fr (L.B.); christophe.riondet@univ-perp.fr (C.R.); 2Laboratoire Génome et Développement des Plantes, CNRS, F-66860 Perpignan, France; 3Centre of Agricultural Biochemistry and Biotechnology, University of Agriculture Faisalabad, 38000 Faisalabad, Pakistan; 4Laboratoire de Spectrométrie de Masse BioOrganique (LSMBO), IPHC, Université de Strasbourg, CNRS UMR 7178, 67037 Strasbourg, France; ccarapito@unistra.fr; 5Institute of Plant Sciences Paris Saclay IPS2, Université Paris-Sud, CNRS, INRA, Université Evry, Paris Diderot, Sorbonne Paris-Cité, Université Paris-Saclay, Bâtiment 630, 91405 Orsay, France; amna.mhamdi@psb.vib-ugent.be (A.M.); graham.noctor@u-psud.fr (G.N.); 6Department of Plant Biotechnology and Bioinformatics, Ghent University, 9052 Gent, Belgium; 7Center for Plant Systems Biology, VIB, 9052 Gent, Belgium

**Keywords:** Isocitrate dehydrogenase, glutathionylation, nitrosylation, glutaredoxin, *Arabidopsis thaliana*

## Abstract

NADP-dependent (Nicotinamide Adénine Dinucléotide Phosphate-dependent) isocitrate dehydrogenases (NADP-ICDH) are metabolic enzymes involved in 2-oxoglutarate biosynthesis, but they also supply cells with NADPH. Different NADP-ICDH genes are found in *Arabidopsis* among which a single gene encodes for a cytosolic ICDH (cICDH) isoform. Here, we show that cICDH is susceptible to oxidation and that several cysteine (Cys) residues are prone to S-nitrosylation upon nitrosoglutathione (GSNO) treatment. Moreover, we identified a single S-glutathionylated cysteine Cys363 by mass-spectrometry analyses. Modeling analyses suggest that Cys363 is not located in the close proximity of the cICDH active site. In addition, mutation of Cys363 consistently does not modify the activity of cICDH. However, it does affect the sensitivity of the enzyme to GSNO, indicating that S-glutathionylation of Cys363 is involved in the inhibition of cICDH activity upon GSNO treatments. We also show that glutaredoxin are able to rescue the GSNO-dependent inhibition of cICDH activity, suggesting that they act as a deglutathionylation system *in vitro*. The glutaredoxin system, conversely to the thioredoxin system, is able to remove S-nitrosothiol adducts from cICDH. Finally, NADP-ICDH activities were decreased both in a *catalase2* mutant and in mutants affected in thiol reduction systems, suggesting a role of the thiol reduction systems to protect NADP-ICDH activities *in planta*. In line with our observations in Arabidopsis, we found that the human recombinant NADP-ICDH activity is also sensitive to oxidation *in vitro*, suggesting that this redox mechanism might be shared by other ICDH isoforms.

## 1. Introduction

Isocitrate dehydrogenases (ICDHs) reversibly catalyze the oxidative decarboxylation of isocitrate to 2-oxoglutarate (2-OG), a key compound in ammonia assimilation by the glutamine synthetase/glutamate synthase pathway. Through their catalytic activity, ICDHs reduce NAD^+^ or NADP^+^, producing NADH or NADPH respectively [1,2]. In plants, both NAD and NADP-dependent ICDH isoforms are found. The NAD-dependent isoform is restricted to mitochondria, where it takes part in the tricarboxylic acid (TCA) cycle [3]. NADP-dependent ICDH isoforms are found in the cytosol, chloroplasts, mitochondria and peroxisomes [4]. In the dicot plant *Arabidopsis thaliana*, a single isoform is found in each cell compartment. The cytosolic isoform (cICDH) is the most abundant form in leaves, as it is responsible for more than 80% of the extractible ICDH activity [5,6]. Biochemical analyses have shown that both mitochondrial ICDH and cytosolic cICDH activities are dependent on the nitrogen status. As both isoforms are involved in nitrogen assimilation, they have been proposed to have overlapping functions [1,2].

NADH and NADPH produced by ICDH are important in reducing equivalents for the regeneration of thiol reduction enzymes like glutathione, thioredoxin or GSNO-reductases [7,8]. Therefore, ICDH have been proposed to play an antioxidant role against oxidative stress, and damage to ICDH may result in the perturbation of the balance between oxidants and antioxidants, and lead to pro-oxidant conditions. This was shown in mammals, where the cytosolic ICDH isoform is highly reactive to peroxynitrite, affecting its activity by formation of nitrotyrosine and S-nitrosothiol adducts [9,10]. Another study in mammalian cells shows that the cytosolic NADP-ICDH activity is regulated by S-glutathionylation [11].

Moreover in mammals, cysteine residues of ICDH play an essential role in the catalytic function of ICDH. Such regulation was not explored in plant isoforms, but evidence suggested that ICDH isoforms can also be redox regulated. Several ICDH isoforms were found to exhibit sulfenylated [12,13], glutathionylated [14] or nitrosylated [15] Cys, suggesting that ICDH cysteines are redox sensitive. Moreover, ICDHs were identified as interactors of thioredoxin (TRX) or glutaredoxin (GRX) in different proteomic approaches [16,17,18,19,20]. Another piece of evidence came from the analyses of a cICDH knock-out mutant in *Arabidopsis*. cICDH protein is dispensable for plant growth and for the leaf basic metabolism. However, *cicdh* mutants exhibit accumulation of defense gene transcripts in the absence of pathogen attack and exacerbate the phenotype and the redox perturbation of the oxidative stress mutant *catalase2* (*cat2*). *cat2* is inactivated in the major leaf catalase isoform, which increases the availability of H_2_O_2_ produced in the peroxisomes [6].

Here, we examine the redox sensitivity of the *Arabidopsis* cytosolic cICDH isoform. We show that enzyme activity is affected by oxidative agents like oxidized glutathione (GSSG) and nitrosoglutathione (GSNO), through modifications of conserved Cys residues in the protein. In particular, Cys363 is S-glutathionylated and can be reversed by the glutaredoxin system. Our data indicate that cytosolic ICDH is redox regulated and that this regulation might be shared in other organisms. 

## 2. Materials and Methods

### 2.1. Plant Materials and Growth Conditions

All *Arabidopsis thaliana* lines used in this study were of Columbia-0 (Col-0) ecotype. The plants were grown in soil in a controlled growth chamber (180 µE m^−2^ s^−1^, 16 h day/8 h night, 22 °C 55% RH day, 20 °C 60% RH night) up to 3 weeks. Plant mutant lines used in this study *ntra ntrb, cat2, gr1, gr1 cat2, icdh, icdh cat2, gsnor, nox1, noa1, nia1 nia2* and *nia1 nia2 noa 1* were previously described [6,21,22,23,24,25,26,27,28].

### 2.2. In Vitro Protein-Based Complementation and TRX/GRX Activity Assays

For *in vitro* protein-based TRX complementation assays, 4.38 μM (200 ng/μL) recombinant cICDH protein was incubated in 25 µL for 2 h on ice with 1 mM NADPH, 4.59 μM TRXh3 or TRXh5 and 3.12 µM NADPH-Dependent Thioredoxin Reductase A (NTRA). This reaction mixture was diluted 40 times in 100 mM phosphate buffer (KOH (Potassium hydroxyde), pH 7.5) and the ICDH activity assay was performed as described above.

For *in vitro* protein-based GRX complementation assays, 4.38 μM (200 ng/μL) recombinant cICDH was incubated in 25 µL for 2 h on ice with 1 mM NADPH, 5 μM GRXC1 or GRXC2, 0.8 mM GSH and 5 µM Glutaredoxin Reductase (GR). This reaction mixture was diluted 40 times in 100 mM phosphate buffer (KOH, pH 7.5) and the ICDH activity assay was performed as described above.

### 2.3. Cloning, Expression and Purification of Recombinant cICDH

The cICDH-coding sequence (At1g65930), with NdeI and BamHI restriction sites at the N- and C-terminal ends was inserted into pET16b vector (Novagen). Point mutation of cysteine 363 to serine was generated using QuikChange II Directed Mutagenesis Kit (Agilent) using primers detailed in Appendix A. The constructs were transferred into *E. coli* BL21 stain and transformed cells were cultured at 37 °C until *A*_600_ = 0.7. cICDH expression was induced by the addition of 1 mM isopropyl-1-thio-d-galactopyranoside (IPTG), followed by further culture at 21 °C for 16 h. All the steps for the purification of recombinant His-tagged cICDH were performed at 4 °C as previously described [29]. Briefly, the cells were resuspended in 50 mM Tris-HCl pH 7.5, disrupted by sonication, and centrifuged at 125,000× *g* for 30 min at 4 °C. The supernatant was loaded onto the Ni^2+^-Sepharose column equilibrated with 50 mM Tris-HCl, pH 7.5, 200 mM NaCl. The obtained fractions containing the purified protein were collected and evaluated on a SDS-PAGE (Sodium dodecyl sulfate-polyacrylamide gel electrophoresis) gel. Purified recombinant protein samples were pooled and stored at 4 °C.

### 2.4. Protein Extracts from Arabidopsis Plants and cICDH Enzymatic Assay

To perform enzymatic assays, Arabidopsis leaf protein extracts were prepared as described before [30,31]. Briefly, 100 mg leaves were ground in a mortar in liquid nitrogen and resuspended in 0.5 mL of a buffer containing 50 mM Hepes/KOH, pH 7.4, 1 mM EDTA, 1 mM EGTA, 2 mM Benzamidine, 2 mM ε-aminocaproic acid, 0.5 mM PMSF, 10% glycerol, 0.1% Triton X-100 and 1 tablet of EDTA-free protease inhibitor cocktail (Roche). After 15 min of incubation on ice, samples were centrifuged at 15,000× *g* and at 4 °C for 15 min to remove tissue debris. NADP-ICDH activity was measured at room temperature from 40 µg of protein extracts in total of 1 mL reaction medium, having 100 mM phosphate buffer (KOH, pH 7.5), 5 mM MgCl_2_, 250 μM of NADP^+^ and 2.5 mM DL-isocitric acid. The activity was monitored as a change in absorbance at 340 nm due to the isocitrate-dependent rate of NADP^+^ reduction. Catalase activity was quantified as described before [6]. Briefly, 20 µg of protein extracts were taken in 50 mM potassium phosphate buffer (pH 7.0) and 10 mM H_2_O_2_ at 25 °C in 1 mL reaction mixture. The activity was monitored as a change in absorbance at 240 nm due to breakdown of H_2_O_2_. To get the reduced cICDH, it was incubated with 20 mM DTT at 25 °C for 1 h. To remove excess dithiothreitol (DTT), protein samples were passed through a micro bio-spin column (Bio-Rad, Marnes-la-Coquette, France) after each treatment. The concentration of modified cICDH measured by absorbance at 340 nm. 

For recombinant cICDH enzymatic assay, the reaction was performed with 800 ng of cICDH at 25 °C in a 1000 µL final volume containing 100 mM phosphate buffer (KOH, pH 7.5), 5 mM MgCl_2_, 250 μM of NADP^+^ and 2.5 mM DL-isocitric acid. The decrease in NADP^+^ absorbance at 340 nm was monitored using a UV-1800 spectrophotometer (Shimadzu, Marne la Vallée, France). The molar extinction coefficient for NADP^+^ of 6220 M^−1^ cm^−1^ was used for the calculation. To obtain the K_M_ for Isocitrate or NADP^+^, progress curves were recorded using varying concentrations of Isocitrate (0–5 mM) or NADP^+^ (0–1 mM). The initial velocity (*vi*) for each substrate concentration was measured, and the *vi*/*E*_0_ values were plotted and fitted with the Michaelis-Menten equation to obtain the kinetic parameters. Three independent replicates of *vi* were measured for each substrate concentration. 

For the reactivation of cICDH by TRXh, cICDH was incubated 30 min with recombinant TRXh (4.59 µM) in the presence of NADPH (0.125 mM) and NTRA (0.5 µM). ICDH activity assay was performed as described earlier after diluting the mix 40-fold. For cICDH reactivation by GRX, it was incubated for 30 min with recombinant GRX (5 µM) in presence of NADPH (0.125 mM), GSH (0.8 mM) and GR (5 µM). Then the mix was diluted 40-fold and used for ICDH activity assay, as described above.

### 2.5. Gel Filtration Chromatography

His-ICDH recombinant proteins (~10 μg) were fractionated using Superose 12 (GE Healthcare, Buc, France) column equilibrated in Protein Buffer (50 mM Tris pH 7.5, 5 mM MgCl_2_) containing 150 mM NaCl. The protein standards were aldolase (158 kDa), conalbumin (75 kDa) and ovalbumin (43 kDa) (GE Healthcare).

### 2.6. Liquid Chromatography-Tandem Mass Spectrometry (LC-MS/MS)

Purified His-ICDH recombinant protein was dissolved in 8 M urea / 0.1 M ammonium bicarbonate buffer and thoroughly vortexed. The urea concentration was then lowered to 1 M by dilution with fresh 0.1 M ammonium bicarbonate, and proteins were digested in solution by addition of trypsin overnight at 37 °C. After acidification with formic acid, desalting and concentration of the peptides were carried out using a C18 Sep-Pak cartridge (Sep-pak Vac 1cc (50mg) tC18 cartridges, Waters, Guyancourt, France). 

LC-MS/MS analyses were conducted on a nanoHPLC-Q Exactive Plus system (Thermo Fisher Scientific, Bremen, Germany). Peptide separation was performed on an ACQUITY UPLC BEH130 C18 column (250 mm × 75 μm with 1.7 μm diameter particles, Waters). The solvent system consisted of 0.1% FA in water (solvent A) and 0.1% FA in ACN (solvent B). Peptides were eluted at 450 nL/min with the following gradient of solvent B: from 1 to 8% over 2 min, from 8 to 35% over 28 min and then 90% for 5 min. The system was operated in positive mode using the following settings: MS1 survey scans (*m*/*z* 300 to 1800) were performed at a resolution of 70 000 with an AGC target of 3 × 10^6^ and the maximum injection time was set to 50 ms. MS/MS spectra were acquired at a resolution of 17500. The MS2 AGC target was set to 1 × 10^5^ and the maximum injection time was set to 100 ms.

NanoLC-MS/MS data were searched using a local Mascot server (version 2.5.1, MatrixScience, London, UK) in an *Arabidopsis thaliana* protein sequences database downloaded from the TAIR site (TAIR10 version, Phoenix Bioinformatics, Fremont, CA, USA), to which decoy sequences were added using the in-house developed software tool MSDA [32]. Spectra were searched with a mass tolerance of 5 ppm in MS and 0.07 Da in MS/MS mode. One missed cleavage was tolerated. Oxidation of cysteine residues were specified as variable modifications. Spectra and fragmentation tables that have allowed identifying glutathionylated peptides are provided.

### 2.7. Biotin Labeling of S-Nitrosylated Proteins

The biotinylation of S-nitrosylated proteins resulting from the *in vitro* S-nitrosylation was detected using the Biotin Switch Test (BST) as described by reference [33] and with minor modifications. To achieve this, 2.19 μM of recombinant wild-type cICDH or Cys-mutated version of cICDH (cICDH-C363S) was incubated with 1 mM GSNO at room temperature in a reaction mixture called HEN buffer. HEN buffer contains protease inhibitor cocktail along with 25 mM 4-(2-hydroxyethyl)-1-piperazineethanesulfonic acid (HEPES) [pH 7.7], 1 mM Ethylenediaminetetraacetic acid (EDTA), and 0.1 mM neocuproine. After 30 min of incubation, the reaction mixture was desalted on Pierce^TM^ Zeba spin columns (Thermo Fisher Scientific, Dardilly, France). To perform the denitrosylation, the resulting protein-SNO were incubated for 45 min with either TRXh, NTRA, and NADPH or with GRXC, GSH, GR and NADPH. Protein S-nitrosylation was assessed using BST as described in reference [33], except for 20 mM NEM being used to alkylate free thiols. Residual NEM was removed by centrifugation (minimum 2000 *g*, 20 min, 4 °C) with 2 volumes of −20 °C acetone (pre-chilled). The supernatant was removed and pellets were resuspended in 0.1 mL of HENS buffer (HEN buffer containing 1% SDS)/mg protein. To achieve biotinylation, the resuspended proteins were incubated at room temperature for 1 h after adding 2 mM biotin-HPDP and 1 mM ascorbate. After removing biotin-HPDP, the precipitated proteins were resuspended in 0.1 mL of HENS buffer/mg of protein and 2 volumes of neutralization buffer (20 mm HEPES, pH 7.7, 100 mm NaCl, 1 mm EDTA, and 0.5% Triton X-100). A total of 15 μL of Streptravidin-agarose/mg of protein were added and incubated for 1 h at RT. The matrix was washed five times with 10 volumes of washing buffer (600 mM NaCl in neutralization buffer). The sample were centrifuged at 200 g for 5 s at room temperature between each wash. Finally, the bound proteins were eluted with 100 mM β-mercaptoethanol in neutralization buffer. 

To perform western blot analysis, SDS-PAGE sample buffer was added to agarose beads. SDS-PAGE gel was run after heating the samples to 70 °C for 10 min. The samples separated on SDS-PAGE were transferred to nitrocellulose membranes. Western blots were probed with anti-His antibodies (Merck KGaA, Darmstadt, Germany).

### 2.8. Bioinformatics Analyses

The gene and protein sequences were obtained from the NCBI website (http://www.ncbi.nlm.nih.gov, Rockville Pike, Bethesda MD, USA). ClustalW2 software (http://www.ebi.ac.uk/Tools/msa/clustalw2/, EMBL-EBI, Wellcome Genome Campus, Hinxton, Cambridgeshire, UK) was used to perform the multiple sequence alignments. Swiss Model software (http://swissmodel.expasy.org/, Protein Structure Bioinformatics Group, Swiss Institute of Bioinformatics Biozentrum, University of Basel, Switzerland) was used for cICDH modeling by using human cytosolic ICDH sequence structure (PDB ID: 1T0L, [34]), and the Geneious 9.0 software (Biomatters ApS, Aarhus, Denmark) was used for the 3D representation.

## 3. Results

### 3.1. Enzymatic Characterization of Cytosolic NADP-ICDH

In order to characterize redox modifications on the cytosolic NADP-ICDH (cICDH, At1g65930), we produced the recombinant cICDH in *E. coli.* The predicted protein shows 77.4% and 85.2% identity with the mito/chloro and peroxisomal NADP-ICDH from *A. thaliana*, respectively. It also shares conserved cysteine residues with most plant and mammalian NADP-ICDH proteins (Appendix A). To verify that cICDH is a genuine NADP-ICDH, the cDNA was cloned in an expression vector and purified using an introduced N-terminal His-tag. SDS/PAGE analysis of purified cICDH indicated a monomer corresponding to the predicated molecular mass of 48.3 kDa calculated for cICDH, including the N-terminal His-tag (Appendix A). The migration of the protein is not modified when the SDS/PAGE gel is run under reducing or non-reducing conditions. However, size exclusion chromatography indicated that purified cICDH eluted at a molecular weight of ~90 kDa, suggesting that the native cICDH is present as a dimer coordinated by non-covalent interactions (Appendix A). Analysis of the catalyzed reaction revealed Michaelis-Menten kinetics with a Km for the substrate isocitrate of 99 µM and a k_cat_ of 4.93 s^−1^ (Figure 1A and Table 1). Km and k_cat_ for NADP^+^ were estimated at 28 µM and and a k_cat_ of 5.81 s^−1^ (Figure 1B). The optimal pH for cICDH activity was found to be in the range of 7.5 to 8.5 (Figure 1C).

Next, we studied the effect of different oxidants on the activity of cICDH. Increasing concentrations of H_2_O_2_ hardly affected the enzyme activity (Figure 2A). However, both increasing concentrations of GSNO and GSSG progressively inhibited the cICDH activity, as did the combination of H_2_O_2_ and GSH (Figure 2). These latter treatments act as GSH donors which might modify cysteine residues by S-glutathionylation. Moreover, GSNO is also an excellent NO donor which can affect cysteine residues by S-nitrosylation. 

We investigated if cICDH can be glutathionylated/nitrosylated by treating His-cICDH protein with GSNO and analyzing samples by mass spectrometry. After trypsin digestion, peptides were analyzed by LC-MS/MS (Figure 3A). This reveals a glutathione adduct on two different Cys363-containing peptides (with and without a missed trypsin cleavage) detected thanks to a 305-Da mass increase of the modified peptides when compared to the non-modified peptide. These experiments attest that Cys363 can be glutathionylated *in vitro*. No other glutathionylated Cys residue was detected. The Cys363 glutathionylated site was confirmed and validated on independent triplicate experiments (Appendix A). Then, we produced a cICDH-C363S mutant protein and tested it for glutathionylation in three same experimental conditions. No glutathionylated residue was detected in three different replicate experiments. Therefore, Cys363 is prone to S-glutathionylation upon GSNO treatment. By modeling cICDH, we established that Cys363 is probably not located in the close proximity of the cICDH active site (Figure 3B).

We also performed a biotin-switch experiment to test if cICDH is S-nitrosylated. Treatment with GSNO leads to a specific biotin-switch signal, indicating that cICDH is S-nitrosylated *in vitro* (Figure 4A). To identify the nitrosylated residues, we subjected the cICDH-C363S mutant protein to the same experiment. No decrease of the biotin-switch signal was observed, which did not allow us to identify the S-nitrosylated Cys residues (Figure 4A).

Then, we examined the denitrosylation activity of thiol reductases by biotin switch (Figure 4B). When GSNO treated cICDH was further incubated by the recombinant cytosolic TRXh3 in the presence or absence of its physiological reducer NTRA, no denitrosylation activity was observed, suggesting that the TRX system was not able to remove SNO adducts from cICDH (Figure 4B). However, adding the GRX system to the reaction triggered the disappearing of the biotin switch signal on GSNO-treated cICDH, suggesting that the GRX-dependent thiol reduction system has a denitrosylation activity on cICDH (Figure 4B). By looking closer to this activity, we noticed that the denitrosylation activity needs the full GRX system (NADPH/GR/GSH/GRX) to be optimal (Figure 4C). Interestingly, while performing the biotin switch in presence of GRXC1, we noticed a strong signal on the GRXC1, suggesting that GRXC1 is prone to trans-nitrosylation (Figure 4B,C).

To test the function of Cys363 in cICDH activity, we examined the impact of the C363S point mutation on enzymatic activities. The cICDH C363S mutant had a Km of 95 µM and a k_cat_ of 4.31 s^−1^ for the substrate isocitrate, which is similar to the wild-type protein, suggesting that the C363S mutation does not perturb the enzymatic characteristics of cICDH (Figure 5A and Table 1). However, cICDH C363S was found to be less affected by GSNO treatment than the wild-type enzyme, suggesting that S-nitrosylation/S-glutathionylation of Cys363 play a regulatory function for cICDH activity (Figure 5B). 

### 3.2. cICDH Activity Is Restored by Glutaredoxins

After having established that cICDH activity is affected by glutathionylation on Cyc363, we wished to determine if cICDH activity could be restored by thiol reduction pathways. For this purpose, we subjected GSNO-treated cICDH to different thiol reductases: thioredoxins are major disulfide reductases, but have also been shown to have denitrosylation capacities [35]. However, they are generally not able to reduce S-glutathionylated cysteine adducts. On the contrary, glutaredoxins exhibit a major deglutathionylation activity [8]. We found that cytosolic GRXC1 and GRXC2 are more efficient than TRXh3 and TRXh5 to restore cICDH activity, which is consistent with a deglutathionylation activity of GRXC1 and GRXC2 (Figure 5C,D). Collectively, our data suggest that cICDH activity is inhibited by S-glutathionylation on Cys363 and that GRXC1 and GRXC2 are able to reverse this effect through their deglutathionylation activity.

### 3.3. Enzyme Activities Are Affected in Mutants

In order to study whether cICDH activity is redox sensitive *in planta*, we measured ICDH activities in leaves from two-weeks old plants of different Arabidopsis mutants. While exhibiting no phenotypic perturbations on the rosette growth, the cytosolic *icdh* KO mutant shows a marked decrease (−60%) of NADP-ICDH activities (Figure 6A). This confirmed previous data showing that the cytosolic ICDH contributed to the major pool of the shoot extractible NADP-ICDH activity. Nevertheless, this contribution seems somehow less than reported previously (−90%), possibly due differences in growth conditions or developmental stage of the plants or to the contribution of organellar NADP-ICDH activities [6]. NADP-dependent ICDH activities were also consistently decreased in the *cat2* mutant (Figure 6A), which is impaired in the major isoform of the peroxisomal H_2_O_2_ detoxification enzyme Catalase 2 (Appendix A), suggesting that NADP-ICDH activities are affected by oxidizing conditions. Interestingly, the steady-state ICDH activity was even more affected (−90%) in the *icdh cat2* double mutant, suggesting that other cellular NADP-ICDH isoforms might be affected by the *cat2* mutation. Moreover, the NADP-ICDH was also slightly decreased in the *gr1* involved in the reduction of the cytosolic glutathione pool [23,36]. The latter mutant accumulates higher levels of oxidized glutathione than wild-type plants, suggesting that NADP-ICDH activities might be sensitive to perturbed glutathione conditions [23]. The NADP-dependent ICDH activities are more strikingly affected in the *cat2 gr1* mutant, which accumulates much higher GSSG levels as *cat2* and *gr1* single mutants [24].

We also examined NADP-ICDH activity in different mutants affected in the NO metabolism [22,26,27,28]. NADP-ICDH activities were decreased in the *gsnor* mutant which accumulates a high level of GSNO. As expected from *in vitro* experiments, treatment of wild-type protein extracts with GSNO strikingly inhibits the activity. Surprisingly, NADP-ICDH steady-state activity is not affected in the *nox1* mutant which overaccumulates NO, suggesting a specific impact of GSNO on ICDH (Figure 6B). This was further confirmed by treating wild-type protein extracts with the NO donor sodium nitroprusside (SNP), which did not affect the NADP-ICDH activities. Finally, the activity is not perturbed in *noa1, nia1 nia2* and *noa1, nia1 nia2* mutants in which the biosynthesis of NO is alleviated (Figure 6B).

## 4. Discussion

Plant ICDH are part of a multigenic family (Appendix A). Mitochondrial NAD-dependent ICDH involved in TCA cycle are composed of 6 genes, two of them are encoding catalytic subunits, and the four others for regulatory subunits [37]. NADP-ICDH are found in the cytosol, peroxisome, mitochondria and chloroplast. Each of these compartments only contains a single isoform, one of them being dually targeted to chloroplasts and mitochondria [4,6,38]. Different arguments are in favor of redox regulation of ICDH: (i) ICDHs were found to exhibit sulfenylated [12,13], glutathionylated [14] or nitrosylated [15] Cys upon oxidizing conditions (ii) ICDH isoforms including the cICDH were previously identified as interactors of thiols reductases TRX and GRX in different proteomic analyses [16,17,18,19,20,39]. (iii) conserved Cys residues are found in ICDH isoforms of most organisms, including mammals (iiii) mammalian cytosolic ICDH have been shown to be regulated by S-nitrosylation [9,10], which we also confirmed in our study (Appendix A). Supporting these assumptions, we found that cytosolic NADP-ICDH activities are affected in GSNO, GSSG and H_2_O_2_/GSH-supplied purified cICDH as well as in mutants exhibiting perturbed GSNO, GSSG or H_2_O_2_ reduction. While these oxidizing conditions seem not to induce intermolecular disulfide bonds in the recombinant enzyme as previously shown for other metabolic enzymes [29], the single Cys363 residue was consistently found to be glutathionylated after GSNO treatments. Interestingly, the Cys363 is also conserved in other ICDH isoforms (mitochondrial, peroxisome and chloroplastic/mitochondrial) as well as in cytosolic ICDH isoforms from other organisms (Appendix A), suggesting that Cys363 redox regulation might occur in other ICDH isoforms. 

Intriguingly, our ICDH model suggests that Cys363 residue is not located close to the ICDH active site and thus seems not to interfere directly with the catalytic activity. A likely hypothesis would be that upon oxidation, a major conformational change occurs, which might change the activity of the enzyme. The fact that the C363S point mutation does not affect the activity of the recombinant cICDH (Figure 5A) does not really support this hypothesis, assuming that a Cys to Ser mutation mimics Cys oxidation. Another hypothesis is that Cys363 glutathionylation (or nitrosylation) occurs that protects the residue from an irreversible overoxidation triggered by oxidative stress [29].

Interestingly, while having no significant impact on cICDH activity, the C363S mutation does affect the sensitivity of the enzyme to GSNO inhibition, which indicates that Cys363 is targeted by GSNO. We also showed that this single mutation does not fully alleviates the sensitivity to GSNO, suggesting that other residues might be involved. Thus, the mechanism by which oxidation affects cICDH activity needs to be further explored. 

It has to be underlined that, while other Cys are found in the cICDH sequence, no other oxidized residues have been identified in our MS/MS experiments under GSNO treatment. Nevertheless, biotin-switch experiments have clearly identified S-nitrosylated residues upon GSNO treatments, suggesting that cICDH is also S-nitrosylated. Whether Cys363 is nitrosylated and if other residues are nitrosylated cannot be concluded from the non-quantitative biotin-switch technique performed on the cICDH-C363S protein and would need additional point mutation experiments. Consistently, an isotope-coded affinity tags (ICAT) approach identified three other S-nitrosylated residues (Cys 75 and Cys269) in cICDH in Arabidopsis [40]. Residue Cys75 of Arabidopsis NADP-ICDH activity has been found to be differentially S-nitrosylated in response to salt stress [41]. And Tyr392 has also been reported to be the only nitrated residue in pea plants and is possibly responsible for the inhibition of catalytic activity following treatment with SIN-1 [42]. These residues are conserved in NADP-ICDH of plants and other organisms (Appendix A). While preparing this manuscript, Munos-Vargas et al. (2018) showed that NO donors peroxynitrite (ONOO^−^), S-nitrosocyteine (CysNO) and DETA-NONOate also inhibited NADP-ICDH activity in sweet pepper. Munos-Vargas et al. (2018) established by *in silico* analysis of the tertiary structure of sweet pepper NADP-ICDH activity (UniProtKB ID A0A2G2Y555) that residues Cys133 and Tyr450 are the most likely potential targets for S-nitrosation and nitration, respectively [43].

Therefore, future mutagenesis experiments should establish the contribution of other residues than Cys363 in the redox regulation of the protein. As S-nitrosylation is a rather unstable modification, the fact that we do not identified S-nitrosylated residues in our LC-MS/MS is not surprising. Interestingly, biotin switch experiments indicate that the TRX system is unable to act as a denitrosylation system for cICDH, as shown for other substrates, including in plants [44,45,46]. In contrast, the GRX system is efficient and needs the complete NADPH/GR/GSH/GRXC1 system to be optimal at least *in vitro*. This is consistent with the observation that the GRXC1 or GRXC2 thiol reduction system is more efficient than the TRX system in rescuing the inhibitory effect of GSNO or GSSG on cICDH activity, suggesting that deglutathionylation activity of GRX is involved in the redox mechanism of cICDH regulation. 

Our NADP-ICDH activity analyses in the *icdh* mutant confirm that cICDH contributes to the major part (~60%) of the NADP-ICDH activity. The other ~40% of the overall extractible activity likely provide from organellar NADP-ICDH activities. These proportions are somehow different than those reported previously, in which cICDH contributed to over 80% of the extractible activity in potato and *Arabidopsis* [5,6]. These differences might be due to different growth conditions (*in vitro* vs soil growth), developmental stages or tissues (10 day-old plantlets vs 3 week-old adult leaves) analyzed. Nevertheless, the observation that the NADP-ICDH activity is more affected in the *cat2 icdh* double mutant compared to the *icdh* single mutant suggests that organellar ICDH isoforms might be affected in by the *cat2* mutation. Nevertheless, we cannot rule out that the decrease in the NADP-ICDH activity in the cat2 mutant also relies on other modifications induced by oxidative distress. 

ICDH is an important enzyme in 2-OG synthesis, as the carbon backbone of ammonia to glutamate by the GS/GOGAT pathway [1]. But it also provides reduced NAD(P)H equivalents to redox enzymes like GR or NTR. Interestingly, both *cat2, gr1* and *cat2 gr1* double mutants exhibit decreased NADPH production [23,25], which might rely on a decreased NADP-ICDH activity. Moreover, the lower ICDH activity we found in the cytosolic *gr1* mutant and to a higher extent in the *cat2 gr1* double mutant is possibly due to the high accumulation of oxidized glutathione found in these mutants. Consistently, the *gsnor* mutant impaired in the reduction of GSNO also exhibits lower ICDH activity, although this effect is less pronounced than an exogenous treatment by GSNO. Intriguingly, both in exogenous treatment with SNP and in the *nox1* mutant which accumulate high levels of NO, the ICDH activity is not affected, suggesting a distinct role of GSNO and NO in ICDH regulation. Such observations were reported previously and suggested that GSNO and NO have distinct substrates [45,46,47]. In the present work, the difference between impact on ICDH activity caused by *gsnor* and *nox1* could result from the fact that GSNO is also an efficient glutathione donor for cICDH S-glutathionylation. 

## 5. Conclusions

Collectively, our data add another evidence of the potential redox regulation of primary carbon and nitrogen metabolism through the regulation of the cytosolic isocitrate dehydrogenase. We also highlight the importance the GRX-dependent thiol reduction systems as a potential actor in fine-tuning the redox regulation of NADP-ICDH activity. While we clearly demonstrate that the enzyme can be modified by glutathionylation *in vitro*, future experiments are needed to demonstrate that this redox regulation actually occurs *in vivo* in response to environmental clues. 

## Figures and Tables

**Figure 1 antioxidants-08-00016-f001:**
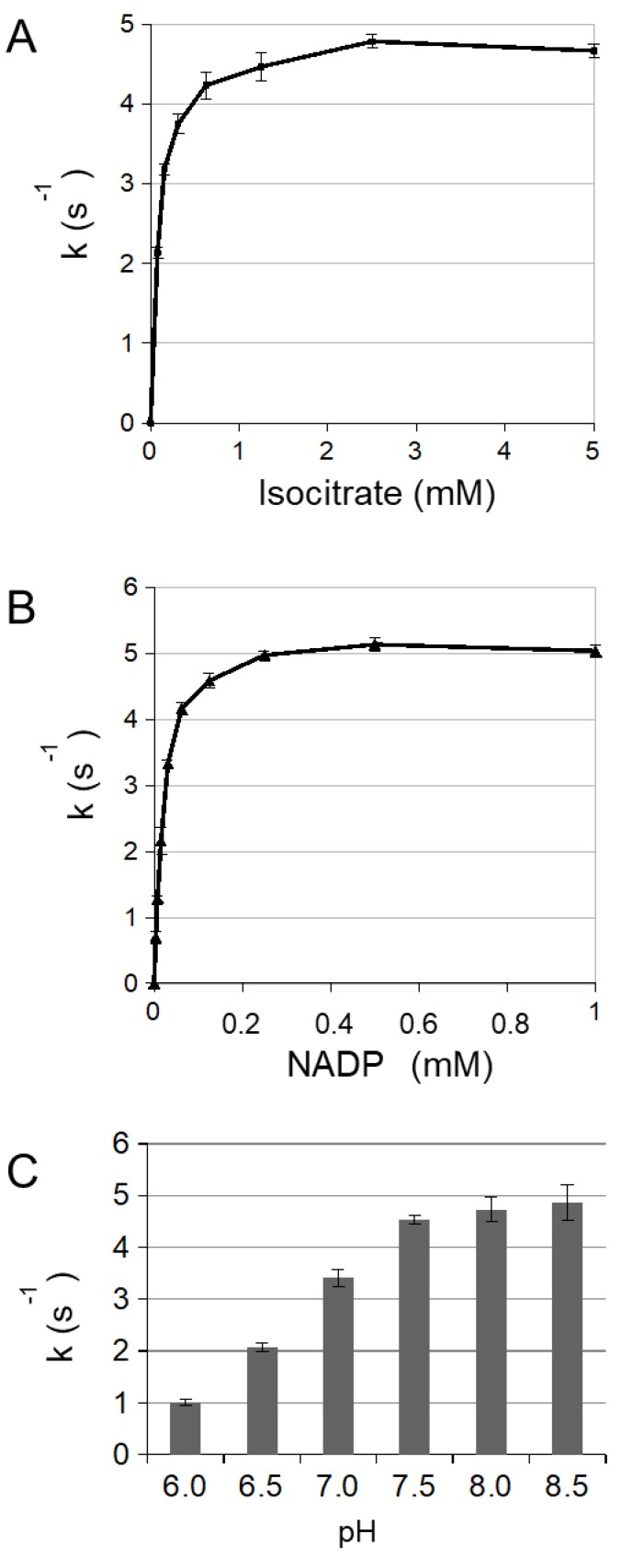
Cytosolic ICDH1 activity. ICDH (Isocitrate dehydrogenase) activity was measured for 800 ng of ICDH by monitoring the change in absorbance at 340 nm due to the isocitrate-dependent rate of NADP^+^ reduction at 25 °C in 1 mL reaction medium containing 5 mM MgCl2, 250 μM of NADP^+^ and (**A**) different concentrations (0–5 mM) of DL-isocitric acid or (**B**) different concentrations (0–1 mM) of NADP in 100 mM phosphate buffer (KOH, pH 7.5). (**C**) 2.5 mM DL-isocitric acid in 100 mM phosphate buffer at different pH (6–8.5). Error bars represent SE (Standard Error) (*n* = 3).

**Figure 2 antioxidants-08-00016-f002:**
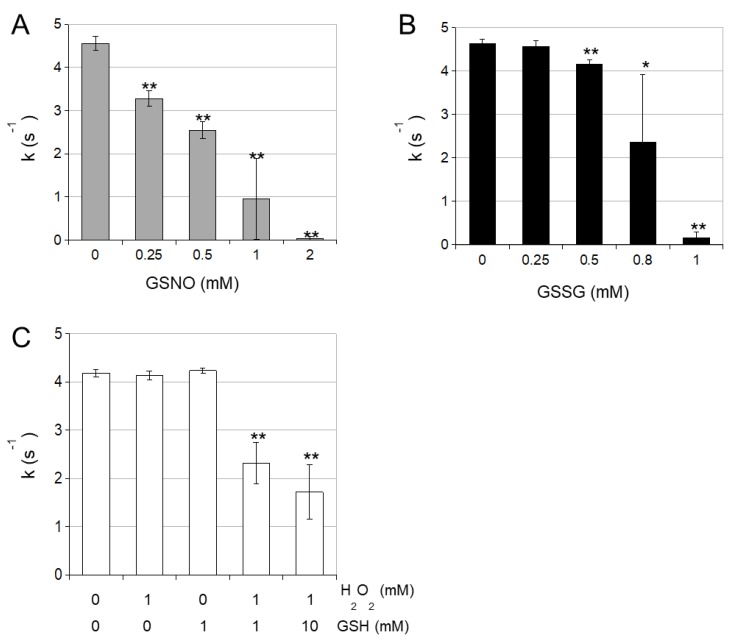
Cytosolic ICDH activity is sensitive to oxidation. 4.38 µM (200 ng/µL) ICDH were incubated with different concentrations of (**A**) H_2_O_2_/GSH, (**B**) nitrosoglutathione (GSNO) or (**C**) oxidized glutathione (GSSG) for 15 min at 25 °C. This reaction mixture was diluted 250 times in reaction buffer during the assay. ICDH activity was measured for 800 ng of ICDH at 25 °C in 1 mL reaction medium containing 100 mM phosphate buffer (KOH, pH 7.5), 5 mM MgCl2, 250 µM of NADP^+^ and 2.5 mM DL-isocitric acid by monitoring the change in absorbance at 340 nm due to the isocitrate-dependent rate of NADP^+^ reduction. Error bars represent SE (*n* = 3). * *p* < 0.05, ** *p* < 0.01 for statistical differences compared to non-treated samples (Student’s *t* test).

**Figure 3 antioxidants-08-00016-f003:**
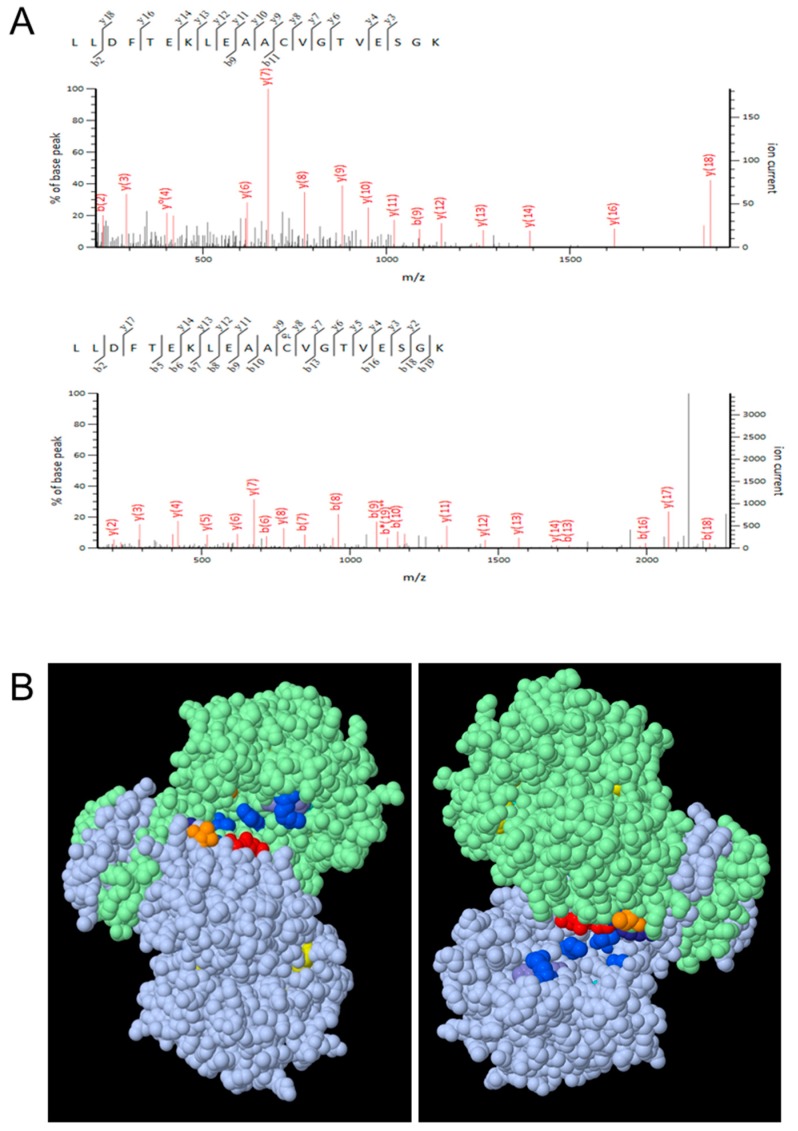
Glutathionylation of cICDH. (**A**) cICDH was treated or not with 1 mM GSNO for 30 min at 25 °C. Samples were trypsin digested and analyzed by nanoLC-MSMS. The panels show fragmentation spectra matching peptides with either unmodified (top) or with glutathionylated C363 (bottom). The same glutathionylated residue was identified in three biological repetitions. (**B**) Modeled cICDH (residues 4–408) homodimer (grey and green chains) based on human cytosolic ICDH. Conserved amino acids (R111, R134, Y141, T214, D252, D279, R314, H315) with the most plant and mammalian NADP-ICDH proteins, in the active site pocket are shown in different colors depending on the nature of the residue: R, blue; Y, purple; T, orange; D, red; H, grey. The conserved cysteine C363 in each chain is represented in yellow (arrowheads). In the right panel, the cICDH homodimer was rotated 180°.

**Figure 4 antioxidants-08-00016-f004:**
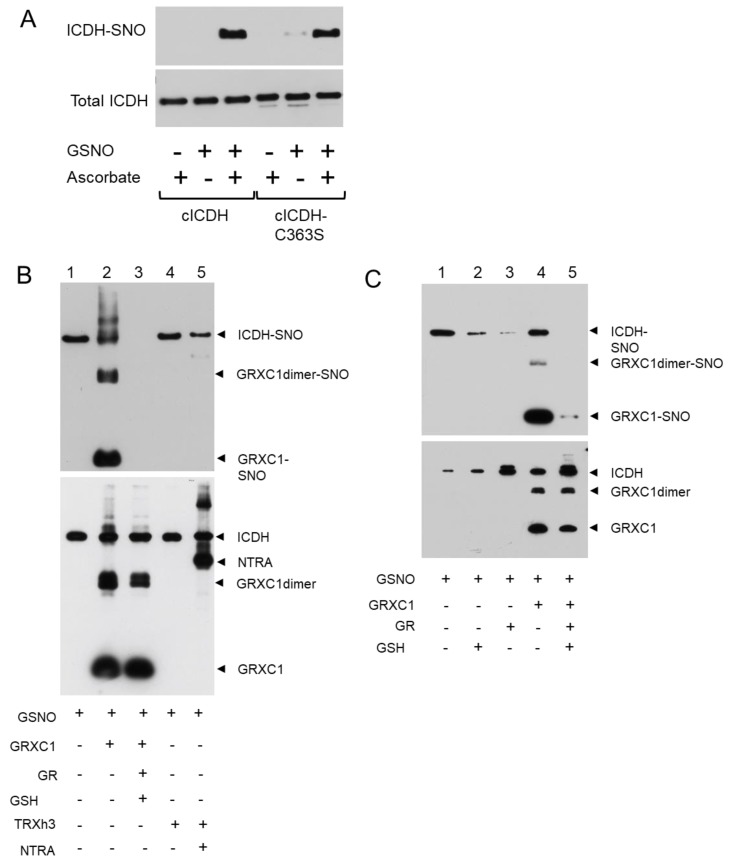
cICDH is S-nitrosylated and is denitrosylated by the GRX system. (**A**) 2.19 µM of recombinant wild-type cICDH or Cys-mutated versions of cICDH (cICDH-C363S) were treated with or without 1 mM GSNO for 30 min at 25 °C, and subjected to the biotin-switch assay in presence or absence of sodium ascorbate. (**B**) After treatment with GSNO (1 mM), the protein was treated with GRXC1 (5 µM) alone (lane 2), GRXC1, GSH (0.8 mM) and GR (0.45 µM) (Lane 3), NTRA (3 µM) alone (lane 4) and NTRA+TRXh3 (4.59 µM) (lane 5) for 30 min at 25 °C and subjected to the biotin-switch assay in the presence of sodium ascorbate. (**C**) The same experimental design in the presence of GSH (0.8 mM) (lane 2), GR (5 µM) (lane 3), GrxC1 (5 µM) (lane 4) and GRXC1+GR+GSH (lane 5). Afterwards, the proteins were separated by reducing SDS-PAGE and transferred onto nitrocellulose membrane. Total ICDH (bottom panel) or S-nitrosylated ICDH (top panel) was detected using an anti-His antibody.

**Figure 5 antioxidants-08-00016-f005:**
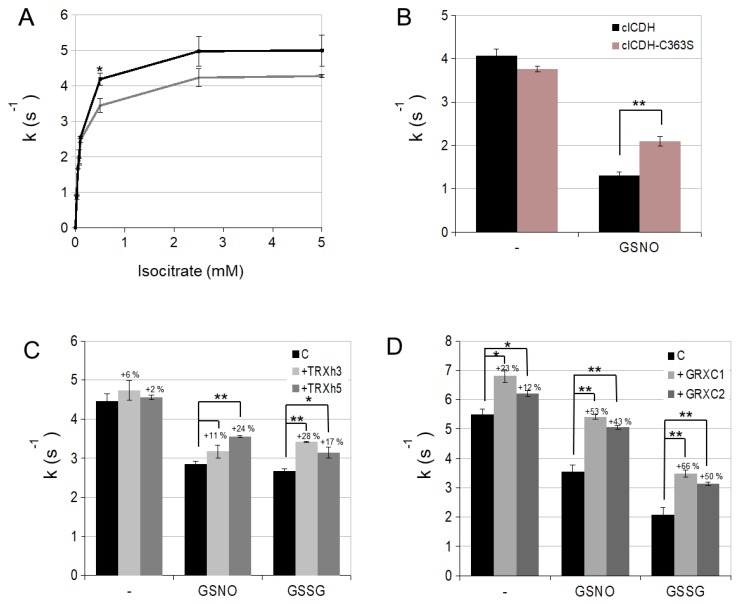
Cysteine-dependent regulation of cICDH activity. (**A**) ICDH activity was measured for 800 ng of ICDH and ICDH-C363S by monitoring the change in absorbance of NADPH at 340 nm due to the isocitrate-dependent rate of NADP^+^ reduction at 25 °C in 1 mL reaction medium containing 5 mM MgCl_2_, 250 μM of NADP^+^ and different (0–5 mM) concentrations of DL-isocitric acid. (**B**) ICDH were incubated with 1 mM GSNO and measured in the same conditions than described previously. (**C**) 800 ng of cICDH were incubated with 0.5 mM GSNO, 0.75 mM GSSG for 15 min at 25 °C. The samples were diluted 2-fold and then incubated with 1 mM NADPH in the presence of NTRA (3 µM) and TRXh3 or TRXh5 (4.59 µM) (**D**) Same experimental design as in (**C**), but in presence of GR (0.45 µM), GSH (0.8 mM) and GRXC1 or GRXC2 (5 µM). Error bars represent SE (*n* = 3). * *p* < 0.01, ** *p* < 0.001 for statistical differences compared to non-treated samples (Student’s *t* test).

**Figure 6 antioxidants-08-00016-f006:**
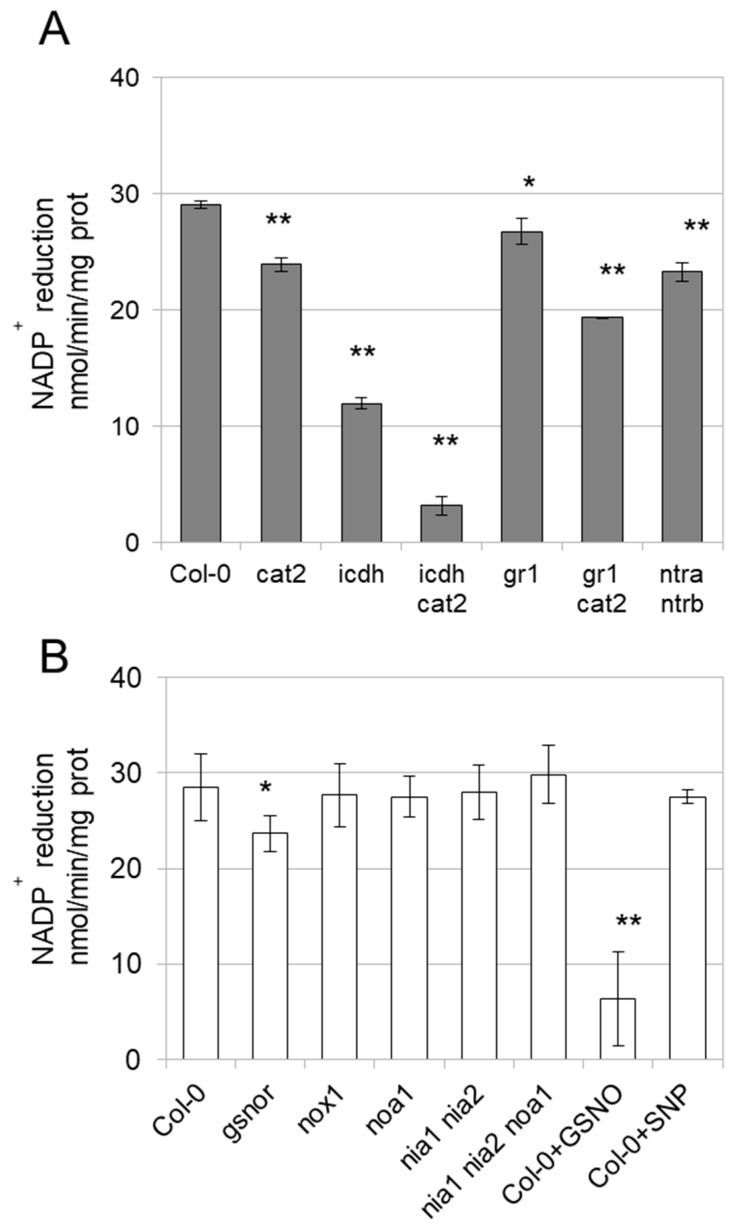
ICDH activity in planta. (**A**) ICDH activity was measured for 40 µg of protein extracts at 25 °C by monitoring the change in absorbance at 340 nm due to the isocitrate-dependent rate of NADP+ reduction. Wild-type (Col-0) or mutant A. thaliana plants were grown in soil in a controlled growth chamber (180 µE m^−2^ s^−1^, 16 h day/8 h night, 22 °C 55% RH day, 20 °C 60% RH night) for 2 weeks. (**B**) Same design as in (A). In Col-O+GSNO and Col-0+SNP, protein extracts of wild-type plants were treated with 1 mM GSNO or SNP for 30 min before performing activity tests. Error bars represent SE (*n* = 3). * *p* < 0.01, ** *p* < 0.001 for statistical differences compared to non-treated samples (Student’s *t* test).

**Table 1 antioxidants-08-00016-t001:** Steady-state enzymatic parameters for cICDH and cICDH-C297S.

Recombinant Protein	*K_M_* Isocitrate (µM)	*k_cat_* Isocitrate (s^−1^)	*k_cat_/K_M_* (M^−1^ s^−1^)
cICDH	99	4.93	4.97 × 10^4^
cICDH-C297S	95	4.31	4.54 × 10^4^

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
