# Peer review of "Cytosolic Isocitrate Dehydrogenase from Arabidopsis thaliana Is Regulated by Glutathionylation"

_antioxidants, 2019, doi:10.3390/antiox8010016_

Round 1

Reviewer 1 Report

Niazi et al. describe a cytosolic, isocitrate dehydrogenase from Arabidopsis, that is apparently modified by glutathionylation at Cys363 after treatment with GSNO. Experiments were carried out to determine the effect of this modification on catalytic activity, and if TRX or GRX causes removal of the modification.  The importance of this form of Cys modification in regulating protein activity under stress conditions is unquestionable, but the data presented here are incomplete, carelessly described, and raise questions such that the story is not ready for publication.  Specific comments are listed below.

Lines 83-4 and 88-9. Treatment of protein “extracts” is described, but neither the sources of those extracts nor the method they were obtained were provided.

Lines 96-7. The His-tag was apparently removed from purified cICDH, but there must be steps to this procedure that were omitted.

Line 100. What is meant by the term “purified cytosolic protein extracts”? Are these pure proteins, or just cytosolic proteins?  How was their purity evaluated?  Neither of the cited papers describe the purification of any protein or even the isolation of cytosolic proteins.  I suspect the authors intended to describe the use of crude leaf extracts that actually contain proteins from all compartments, not just the cytosol.  As such, the results obtained should be interpreted with caution because they likely contained all of the forms of ICDH, not just the cytosolic form.

Line 126. Gel filtration standards are mentioned, but only one peak representing cICDH is shown in Figure S3. How is the reader to know how the standards were used?

Line 188. It likely is not necessary to purify His-tagged proteins to “homogeneity” in order to use them in biochemical assays, but there is no evidence provided here to justify the use of this term. However, the SDS gel shown in Figure S2 reveals that the purified proteins were not homogeneous, and thus using them in MS experiments is questionable.

Lines 206-210. This is background information better placed in the introduction. 

Line 217. The authors claim that no other cysteines were found to be modified, but they don’t claim to have identified all of the Cysteine residues in their MS data.

Line 221. MS data on WT and Cys363S proteins are used to claim that Cys363 is glutathionylated.  This would be more convincing if anti-GSH western blots were carried out on both proteins.

Figure 3B shows a homology model of the protein in question, but this figure is not referenced in the text.  Moreover, the conserved residues in the active site are colored by residue type, but the colors used are not associated with the types.  What does the “conservation” refer to – orthologs, paralogs, all plants, all organisms?

Figure 4.  “A” is not in the legend, and a double mutant mentioned in the legend (Cys297/363S) is not mentioned anywhere in the paper.  In Figure 4A it seems that the lanes are mislabeled since there is no band in the GSNO lane when there should be, and there is a band in the GSNO + ascorbate lanes where there should not be.  Importantly, there is no difference between the WT and mutant proteins.  This clearly reveals that Cys363 is not the only residue modified by GSNO and throws into question the validity of the entire story.

Line 241.  Here the kinetic parameter “Vmax” is used, but in line 199 “kcat” is used.  Please be consistent. 

Line 248.  The interchangeable use of glutathionylation and nitrosylation to describe the modification of Cys363 is confusing.

Figure 5B. The impact of the C363S mutation on sensitivity of cICDH to inhibition by GSNO indicates that this cysteine is not the only one that when modified causes the reduction of activity. Mutating all of the Cys residues would lead to a much clearer picture of the effect of Cys modification on this enzyme.

Line 244. It does appear that the C363S mutant is less sensitive to inhibition by GSNO, but to say that it is “much less affected” is an overstatement. 

Line 254. The description that TRXh is “hardly able to restore” activity while various GRX forms are “able to restore activity” is misleading.  In both experiments, there is some recovery and the differences are not that large. Moreover, some of the treatments have the same effect in the absence of GSNO.  In addition, in the legend, descriptions of experiments in C and D are reversed.

Line 261.  The term “plantlets” is vague, and the statement that there were no phenotypic perturbations is meaningless without a description of what was examined. 

Line 278.  These experiments are rather confusing.  The fact that activity was only slightly lower in gsnor and not at all lower in nox1 suggest that in vivo activity is not significantly affected by redox modification of the cICDH.  Perhaps a treatment could be used that should lead to accumulation of NO or GSNO, then comparison of the effect in WT and mutant plants that are unable to make NO would be revealing.

Figure S1. This sequence alignment is difficult to use and contains misleading details.  The numbering system makes it difficult to identify individual residues by number (e.g. Cys363).  Also, some serine residues are highlighted with yellow, but in the legend yellow is stated to indicate a conserved cysteine.  How were the conserved, active site residues identified? Also, sequences should be identified in the legend by their specific gene number.

At least one important reference was not included: Shin et al. (2009) Glutathionylation regulates cytosolic NADP+-dependent isocitrate dehydrogenase activity. Free Radic Res. 2009 Apr;43(4):409-16. doi: 10.1080/10715760902801525.

Author Response

Niazi et al. describe a cytosolic, isocitrate dehydrogenase from Arabidopsis, that is apparently modified by glutathionylation at Cys363 after treatment with GSNO. Experiments were carried out to determine the effect of this modification on catalytic activity, and if TRX or GRX causes removal of the modification.  The importance of this form of Cys modification in regulating protein activity under stress conditions is unquestionable, but the data presented here are incomplete, carelessly described, and raise questions such that the story is not ready for publication.  Specific comments are listed below.

Lines 83-4 and 88-9. Treatment of protein “extracts” is described, but neither the sources of those extracts nor the method they were obtained were provided.

Answer : Right. We have corrected the mistake in the Materials and Methods. It is not protein extracts but recombinant cICDH.

Lines 96-7. The His-tag was apparently removed from purified cICDH, but there must be steps to this procedure that were omitted.

Answer : We apologize for the mistake. Here, the protein was not cleaved from its His-Tag peptide. We modified the sentence in line 96 accordingly.

Line 100. What is meant by the term “purified cytosolic protein extracts”? Are these pure proteins, or just cytosolic proteins? How was their purity evaluated? Neither of the cited papers describe the purification of any protein or even the isolation of cytosolic proteins. I suspect the authors intended to describe the use of crude leaf extracts that actually contain proteins from all compartments, not just the cytosol. As such, the results obtained should be interpreted with caution because they likely contained all of the forms of ICDH, not just the cytosolic form.

Answer : We apologize for the approximation in the Materials and Methods. It is true that the extracts are crude leaf extracts and that they can contain proteins from other compartments than the cytosol. We have modified the Materials and Methods by adding details on the preparation of these extracts. Indeed, the residual NADP-ICDH activity which was detected in the icdh mutant (~40% of Wild-type) might provide from organellar activities. We have now taken in account this point and discussed it properly in the results and discussion parts.

Line 126. Gel filtration standards are mentioned, but only one peak representing cICDH is shown in Figure S3. How is the reader to know how the standards were used?

Answer : Right, we have now added the standard curve in Figure S3.

Line 188. It likely is not necessary to purify His-tagged proteins to “homogeneity” in order to use them in biochemical assays, but there is no evidence provided here to justify the use of this term. However, the SDS gel shown in Figure S2 reveals that the purified proteins were not homogeneous, and thus using them in MS experiments is questionable.

Answer : We agree with the reviewer comment and have removed the term « homogeneity ». It is true that gel in Figure S2 shown some minor bands that reveal some contaminants in the purified fractions. We were not able to improve the purification design to avoid these contaminants. Nevertheless, we do not think that this question our MS/MS data, since it did not interfere with peptide identification by our bioinformatics softwares.

Lines 206-210. This is background information better placed in the introduction. 

Answer : Agree, we moved this part to the introduction

Line 217. The authors claim that no other cysteines were found to be modified, but they don’t claim to have identified all of the Cysteine residues in their MS data.

Answer : Indeed, this is general caution that needs to be taken when exploiting and discussing MS data. Claims and conclusions can only be drawn about peptides that have been detected and not about peptides that could not be detected. We have identified and validated peptides containing glutathionylated Cys363 in independent triplicate experiments and this is the reason why we have claimed that Cys363 can be glutathionylated in vitro. Besides that, we could not identify any other cysteine residue in a glutathionylated form. Though, having identified a cysteine residue in a non-modified form does not give us enough arguments to claim that it can absolutely not be modified, as it could be glutathionylated at very low levels that would lead to signals below the detection limits of our current experimental setup. We therefore really would like to keep the sentences as formulated in the present version of the manuscript, without overclaiming about the status of other cysteine residues: “These experiments attest that Cys363 can be glutathionylated in vitro. No other glutathionylated Cys residue was detected. Cys363 glutathionylated site was confirmed and validated on independent triplicate experiments (Supplemental Fig. 4).”

Line 221. MS data on WT and Cys363S proteins are used to claim that Cys363 is glutathionylated.  This would be more convincing if anti-GSH western blots were carried out on both proteins.

Answer : We agree with the reviewer that performing an anti-GSH western blot would add another evidence for glutathionylation of Cys363. Nevertheless, we believe that our MS/MS are solid enough to conclude that Cys 363 is indeed glutathionylated.

Figure 3B shows a homology model of the protein in question, but this figure is not referenced in the text. Moreover, the conserved residues in the active site are colored by residue type, but the colors used are not associated with the types. What does the “conservation” refer to – orthologs, paralogs, all plants, all organisms?

Answer : Agree, we have now refered to the figure in the text. We have also modified the figure legend to associate the residue colors with the type of amino acids. We have also better specified the term « conservation » in the text, which refer to orthologs in all organisms.

Figure 4.  “A” is not in the legend, and a double mutant mentioned in the legend (Cys297/363S) is not mentioned anywhere in the paper. In Figure 4A it seems that the lanes are mislabeled since there is no band in the GSNO lane when there should be, and there is a band in the GSNO + ascorbate lanes where there should not be.  Importantly, there is no difference between the WT and mutant proteins.  This clearly reveals that Cys363 is not the only residue modified by GSNO and throws into question the validity of the entire story.

Answer : We apologize for the mistake in the legend. We added the « A » in the legend and removed Cys297/363S. Indeed, while trying to identify additional nitrosylated Cys, we mutated Cys297 to Ser and performed the biotin-switch with the Cys297/363S double mutant protein. Indeed, the results were similar than the Cys363S. As this construct did not add any information on teh targeted Cys, we decided not to present the double mutant in the manuscript. We do not agree with the reviewer concerning the limited role of Cys363 in this story. Although, we show that Cys363 is not the only S-nitrosylated Cys in the protein, we do show in figure 5B that it significantly changes the sensitivity of the Cys363S-ICDH to GSNO treatment. However, we do not claim that Cys363 has an exclusive role in the regulation. This statement is clearly exposed in the discussion (lines 406-410).

There is no mislabeled lanes in Figure 4A. The biotin-switch signal is only revealed by adding ascorbate (which reduce -SNO adducts) to the samples. Minus Ascorbate samples are only used as control to show that the signal is specific to SNO adducts.The reviewer were probably misleaded by Figure 4B and C, where all samples are treated by ascorbate in the biotin-switch test. We have added this point in the figure legend.

Line 241.  Here the kinetic parameter “Vmax” is used, but in line 199 “kcat” is used. Please be consistent. 

Answer : Thank you for the remark, we changed this in the text.

Line 248. The interchangeable use of glutathionylation and nitrosylation to describe the modification of Cys363 is confusing.

Answer : Right, we also modified this in the text.

Figure 5B. The impact of the C363S mutation on sensitivity of cICDH to inhibition by GSNO indicates that this cysteine is not the only one that when modified causes the reduction of activity. Mutating all of the Cys residues would lead to a much clearer picture of the effect of Cys modification on this enzyme.

Answer : Agree, but this would need much more experiments which cannot be made in the short term of the resubmission process. But our experiments already indicate that the single Cys363 plays a important role for the redox sensitivity of the protein as the mutation renders the protein partially insensitive to GSNO treatment. This is well supporting the hypothesis that Cyc363  glutathionylated is regulating the NADP-ICDH activity. However, the fact that other Cys are likely S-nitrosylated (Fig4) also suggests that other Cys are involved. As indicated above, we have started to explore the role of other Cys by associating point mutation of both Cys363S and Cys297S. As the additional Cys297 mutation did not have a significant impact on the activity, we decided not to include it to this manuscript.

Line 244. It does appear that the C363S mutant is less sensitive to inhibition by GSNO, but to say that it is “much less affected” is an overstatement. 

Answer : Agree, we modified this over statement by removing the term « much ».

Line 254. The description that TRXh is “hardly able to restore” activity while various GRX forms are “able to restore activity” is misleading. In both experiments, there is some recovery and the differences are not that large. Moreover, some of the treatments have the same effect in the absence of GSNO. In addition, in the legend, descriptions of experiments in C and D are reversed.

Answer : Regarding the pourcentage of recovery, we are confident that GRX has a more efficient impact on the recovery than TRX. Nevertheless, we agree with the reviewer comment and have amended the text accordingly. We also corrected the legend.

Line 261.  The term “plantlets” is vague, and the statement that there were no phenotypic perturbations is meaningless without a description of what was examined. 

Answer : Agree, we have added more details in the text.

Line 278. These experiments are rather confusing. The fact that activity was only slightly lower in gsnor and not at all lower in nox1 suggest that in vivo activity is not significantly affected by redox modification of the cICDH. Perhaps a treatment could be used that should lead to accumulation of NO or GSNO, then comparison of the effect in WT and mutant plants that are unable to make NO would be revealing.

Answer : To our opinion, the fact that NADP-ICDH activity is only slightly lower in the gsnor mutant is not really surprizing. We did not expect to monitor a full inactivation of the enzyme. That is the reason why we also included in the experiment an exogenous treatment of wild-type plant extract with GSNO, in which we expected a much stronger effect and which is indeed observed. The lack of effect of the nox1 mutation is more surprizing but this difference has been already described in the litterature and we refered to it in the discussion. It is also possible that the exclusive decrease of the NADP-ICDH activity in the gsnor is indeed due to S-glutathionylation of cICDH by GSNO and not by S-nitrosylation. We followed the suggestion from the referee and added the treatment of wild-type plant extracts with the NO donor SNP. This treatment did not significantly affect the activity (Fig 6), which seems to confirm the data obtained in the nox1 mutant. We further discussed the different points in the text.

Figure S1. This sequence alignment is difficult to use and contains misleading details. The numbering system makes it difficult to identify individual residues by number (e.g. Cys363).  Also, some serine residues are highlighted with yellow, but in the legend yellow is stated to indicate a conserved cysteine. How were the conserved, active site residues identified? Also, sequences should be identified in the legend by their specific gene number.

Answer : We agree with this comment and have modified the figure and legend accordingly : (1) we improved the quality of the figure (2) we removed yellow labeling of Ser residues. (3) we refered to references which were used to determine the conserved active site residues (4) we also add details on the numbering of the sequence and highlighted the Cys363 residue (5) we added gene numbers in the legend.

At least one important reference was not included: Shin et al. (2009) Glutathionylation regulates cytosolic NADP+-dependent isocitrate dehydrogenase activity. Free Radic Res. 2009 Apr;43(4):409-16. doi: 10.1080/10715760902801525.

Answer : Thank you for pointing this oversight. We have added this reference in the text.

Reviewer 2 Report

Niazi et. al., investigated the regulation of recombinant cytosolic isocitrate dehydrogenase activity by glutathionylation. The authors found that the recombinant cICDH is glutathionylated at Cys363 residue and is inhibited the enzymatic activity of cICDH. Furthermore, they showed that the glutathionylation and nitrosylation of cICDH are regulated by Grx/Trx system. The experiments are well designed and the manuscript is written clearly. I only have some minor comments on clarify of writing and recommend this manuscript to pubulish.

In figure 1, the authors should be show the optimal temperature for enzymatic activity. It will help us to understand the enzymatic characteristics of cICDH.

If possible, It is good to show the enzyme kinetics parameter including Vmax, Km, Kcat at the table.

Author Response

Niazi et. al., investigated the regulation of recombinant cytosolic isocitrate dehydrogenase activity by glutathionylation. The authors found that the recombinant cICDH is glutathionylated at Cys363 residue and is inhibited the enzymatic activity of cICDH. Furthermore, they showed that the glutathionylation and nitrosylation of cICDH are regulated by Grx/Trx system. The experiments are well designed and the manuscript is written clearly. I only have some minor comments on clarify of writing and recommend this manuscript to pubulish.

Answer :Thank you for the positive evaluation.

In figure 1, the authors should be show the optimal temperature for enzymatic activity. It will help us to understand the enzymatic characteristics of cICDH.

Answer : We have added in the figure legend the optimal temperature used in all enzymatic tests, which is 25°C.

If possible, It is good to show the enzyme kinetics parameter including Vmax, Km, Kcat at the table.

Answer :We have included as a new Table1 the kinetic parameters of the enzyme.

p { margin-bottom: 0.25cm; line-height: 115%; }

Reviewer 3 Report

Here, Niazi et al tested if cytosolic isocitrate dehydrogenase from Arabidopsis plant could be subjected to S-glutathionylation and S-nitrosylation. The authors found that purified ICDH can be subjected to glutathionylation which can be reversed by glutaredoxins. In addition, it was found that modifying ICDH with glutathione lowers the activity of the enzyme. Finally, the authors demonstrated that ICDH activity is affected in plants harbouring a mutation in catalase.

Overall, the study is of interest and the experiments and methods used were appropriately implemented to investigate the glutathionylation state of ICDH. My chief concerns are as follows;

First the authors were able to show that ICDH can be modified in vitro. This hardly qualifies a mechanism for redox regulation since there is no data demonstrating that it occurs in vivo in response to environmental cues. It is strongly recommended that the authors amend the manuscript to reflect the fact that they were only able to show that ICDH can be modified with glutathione.

My second concern involves figure 6, which could have been a good opportunity to measure the glutathionylation state of ICDH in plants. My concern stems from the observation that loss of catalase or gr1 is associated with a decrease in NADPH production. Unfortunately the authors provide no evidence that the decrease in production is related to the glutathionylation of ICDH. It is far more likely that the decrease is related to a range of redox modifications, which can occur non-specifically during oxidative distress. Does loss of catalase cause oxidative distress? Can ICDH be subjected to other types of modifications following oxidative distress (e.g. lipidation by lipid hydroperoxide breakdown products, sulfenylation, sulfonamide formation et cetera)?

Author Response

Here, Niazi et al tested if cytosolic isocitrate dehydrogenase from Arabidopsis plant could be subjected to S-glutathionylation and S-nitrosylation. The authors found that purified ICDH can be subjected to glutathionylation which can be reversed by glutaredoxins. In addition, it was found that modifying ICDH with glutathione lowers the activity of the enzyme. Finally, the authors demonstrated that ICDH activity is affected in plants harbouring a mutation in catalase.

Overall, the study is of interest and the experiments and methods used were appropriately implemented to investigate the glutathionylation state of ICDH. My chief concerns are as follows;

First the authors were able to show that ICDH can be modified in vitro. This hardly qualifies a mechanism for redox regulation since there is no data demonstrating that it occurs in vivo in response to environmental cues. It is strongly recommended that the authors amend the manuscript to reflect the fact that they were only able to show that ICDH can be modified with glutathione.

Answer : We are fully agree with this important remark. Nevertheless, while our work does not demonstrate an in vivo occurance of the ICDH glutathionylation, it opens the way for future experiments. Our experiments showing contrasted NADP-ICDH activitiesin mutants is already informative.Recent worksfrom the litteraturesuggesting thatNADP-ICDH activity is modulated by Cys modifications during stress conditions and fruit maturation goes also in this way. We have amended the abstract and the discussion to clarify thesepoints.

My second concern involves figure 6, which could have been a good opportunity to measure the glutathionylation state of ICDH in plants. My concern stems from the observation that loss of catalase or gr1 is associated with a decrease in NADPH production. Unfortunately the authors provide no evidence that the decrease in production is related to the glutathionylation of ICDH. It is far more likely that the decrease is related to a range of redox modifications, which can occur non-specifically during oxidative distress. Does loss of catalase cause oxidative distress? Can ICDH be subjected to other types of modifications following oxidative distress (e.g. lipidation by lipid hydroperoxide breakdown products, sulfenylation, sulfonamide formation et cetera)?

Answer :Thanks for pointing the decrease of NADPH production in the cat and gr1 mutants, which we did not discussed previously. We have added this point in the discussion. We agree that providing evidence that the decrease in NADPH production is related to ICDH glutathionylation would have been very interesting. However, these experiments are not easy to perform inplanta. For example, we would need to perform OxICAT experiments in our mutants, which is to our knowledge very difficult to do in planta, due to low reproducibility of the data. Another option would be to immunoprecipitate ICDH from protein extracts and detect glutathionylation using anti-GSH antibodies. However, we do not have specific antibodies against the cytosolic NADP-ICDH, which limit this appoach.

The cat2 mutant grown under our conditions (two weeks in soil) startsto show some visible bleached lesions, suggesting that it exhibits oxidative distress. We agreethatwe cannot rule out that the decreasein the NADP-ICDH activity in the cat2 mutant rely onother modifications induced by oxidative distress.Althoughoxidative distress-associated modifications of ICDH has not been documented so far, we have added this hypothesis in the discussion.

p { margin-bottom: 0.25cm; line-height: 115%; }

Round 2

Reviewer 1 Report

Niazi et al. are to be commended for significantly improving the manuscript and addressing all of the comments I made.  I apologize for my mistake regarding the labeling of Figure 4A. 

Several minor points are listed below.

Line 2.  The title implies that the glutathionylation occurs in vivo, but this was not addressed.  I strongly recommend changing the title to “Cytosolic Isocitrate Dehydrogenase from Arabidopsis thaliana is Regulated by Glutathionylation.”

Line 105 (of the track changes version).  This section begins with “The constructs…” but the source and nature of the constructs have not been introduced or described at this point.

Line 118.  The word “grinded” should be “ground.”

Line 127.  The reference “Beers and Sizer, 1952” should be numbered here and isn’t listed in the references.

Line 144.  The word “it” has no reference.  Later the phrase “half an hour” should probably be 30 min. 

Line 212-3.  The line “Downstream bioinformatic analysis..” is unnecessary.

Figure 3B.  The changes to the legend are fine, but the location of the cysteine is not obvious to the reader, especially in the right panel.  Perhaps arrows pointing the location of the cysteine residue in each monomer would help.

Line 300-1.  Because the biotin-stitch signal was unaffected in the C363S mutant, doesn’t this indicate that C363 cannot be the nitrosylated residue?  If so, the stated conclusion can be strengthened.  This evidence should also be mentioned in the Discussion, probably in line 412.

Line 343.  The word “are” should be “is.”

Line 464.  The word “provide” isn’t the right word.  Do the authors mean “result”?

Author Response

Niazi et al. are to be commended for significantly improving the manuscript and addressing all of the comments I made.  I apologize for my mistake regarding the labeling of Figure 4A. 

Answer : Thank you for your positive evaluation.

Several minor points are listed below.

Line 2.  The title implies that the glutathionylation occurs in vivo, but this was not addressed.  I strongly recommend changing the title to “Cytosolic Isocitrate Dehydrogenase from Arabidopsis thaliana is Regulated by Glutathionylation.”

Answer : Right, we changed the title according to your comment.

Line 105 (of the track changes version).  This section begins with “The constructs…” but the source and nature of the constructs have not been introduced or described at this point.

Answer : Right, we added these informations and also detailed on primers (additional Supplemental Table 1) used for cloning and mutagenesis of Cys.

Line 118.  The word “grinded” should be “ground.”

Answer : Corrected.

Line 127.  The reference “Beers and Sizer, 1952” should be numbered here and isn’t listed in the references.

Answer : Corrected.

Line 144.  The word “it” has no reference.  Later the phrase “half an hour” should probably be 30 min. 

Answer : Corrected.

Line 212-3.  The line “Downstream bioinformatic analysis..” is unnecessary.

Answer : Corrected.

Figure 3B.  The changes to the legend are fine, but the location of the cysteine is not obvious to the reader, especially in the right panel.  Perhaps arrows pointing the location of the cysteine residue in each monomer would help.

Answer : Thank you for the comment. We have modified the figure accordingly.

Line 300-1.  Because the biotin-stitch signal was unaffected in the C363S mutant, doesn’t this indicate that C363 cannot be the nitrosylated residue?  If so, the stated conclusion can be strengthened.  This evidence should also be mentioned in the Discussion, probably in line 412.

Answer : We consider that thebiotin-switch assay is not quantitative enough to conclude that C363S is not S-nitrosylated. So, we kept our conclusion but added a sentence in the text to justifythis point.

Line 343.  The word “are” should be “is.”

Answer : Corrected.

Line 464.  The word “provide” isn’t the right word.  Do the authors mean “result”?

Answer : Corrected.

p { margin-bottom: 0.25cm; line-height: 115%; }